# Isolation of *Salvia miltiorrhiza* Kaurene Synthase-like (*KSL)* Gene Promoter and Its Regulation by Ethephon and Yeast Extract

**DOI:** 10.3390/genes14010054

**Published:** 2022-12-24

**Authors:** Piotr Szymczyk, Łukasz Kuźma, Agnieszka Jeleń, Ewa Balcerczak, Małgorzata Majewska

**Affiliations:** 1Department of Biology and Pharmaceutical Botany, Medical University of Łódź, Muszyńskiego 1, 90-151 Łódź, Poland; 2Department of Pharmaceutical Biochemistry and Molecular Diagnostics, Medical University of Łódź, Muszyńskiego 1, 90-151 Łódź, Poland

**Keywords:** callus culture, promoter, *cis*-active element, tanshinone, ethylene, yeast extract

## Abstract

The presented study describes the regulation of the promoter region of the *Salvia miltiorrhiza* kaurene synthase-like gene (*SmKSL*) by ethylene and yeast extract. The isolated fragment is 897 bp and is composed of a promoter (763 bp), 5′UTR (109 bp), and a short CDS (25 bp). The initial in silico analysis revealed the presence of numerous putative *cis-*active sites for *trans*-factors responding to different stress conditions. However, this study examines the influence of ethylene and yeast extract on *SmKSL* gene expression and tanshinone biosynthesis regulation. The results of 72h RT-PCR indicate an antagonistic interaction between ethylene, provided as ethephon (0.05, 0.10, 0.25, and 0.50 mM), and yeast extract (0.5%) on *SmKSL* gene expression in callus cultures of *S. miltiorrhiza*. A similar antagonistic effect was observed on total tanshinone concentration for up to 60 days. Ethylene provided as ethephon (0.05, 0.10, 0.25, and 0.50 mM) is a weak inducer of total tanshinone biosynthesis, increasing them only up to the maximum value of 0.67 ± 0.04 mg g^−1^ DW (60-day induction with 0.50 mM ethephon). Among the tanshinones elicited by ethephon, cryptotanshinone (52.21%) dominates, followed by dihydrotanshinone (45.00%) and tanshinone IIA (3.79%). In contrast, the 0.5% yeast extract strongly increases the total tanshinone concentration up to a maximum value of 13.30 ± 1.09 mg g^−1^ DW, observed after 50 days of induction. Yeast extract and ethylene appear to activate different fragments of the tanshinone biosynthesis route; hence the primary tanshinones induced by yeast extract were cryptotanshinone (81.42%), followed by dihydrotanshinone (17.06%) and tanshinone IIA (1.52%).

## 1. Introduction

*S. miltiorrhiza* Bunge holds a distinguished position among other medicinal plants since it is a traditional source of medically-active tanshinones. The medicinal properties of tanshinones have been exploited in China since ancient times to treat chronic heart failure [1]. The clinical applications of *S. miltiorrhiza* have recently broadened to include neuropathic pain, hyperlipidemia, alcoholism, Parkinson’s, and Alzheimer’s disease [2,3,4]. Its popularity in research can be attributed to its relatively short life cycle, ease of propagation, and availability of genome transformation systems. In addition, it has a range of well-developed in vitro culture systems and a relatively small draft genome sequence of about 538 MB, containing 30,478 predicted genes [5]. Tanshinones are predominantly acquired from the roots of *S. miltiorrhiza* field plants. However, this source is becoming steadily impoverished by climate change, soil pollution, water deficit, and incremental stepping of arable area, making traditional production more difficult [6]. The most efficient alternative sources of *S. miltiorrhiza* plant material are hairy roots and calli grown on solid medium or as suspension callus cell cultures [7,8,9]. These cultures provide a continuous, season-independent supply of plant material. Moreover, the potential conversion of callus cultures into suspension cell cultures could be important for industrial-scale production in bioreactors [10]. These suspension cell cultures demonstrate considerable similarity to the animal cell cultures broadly used as leading sources of recombinant proteins for medicine and industry [11,12]. As such, industrial approaches for processscale-up, media, or process optimization could be applied to plant cell cultures. Examples of such successful plant suspension cultures methods include the production of taliglucerase α (Elelyso) in carrot (*Daucus carota*) used to treat Goucher syndrome and Paclitaxel from yew (*Taxus* sp.) applied as a cytostatic factor in cancer chemotherapy [13,14]. Callus culture offers great potential as a tool for manufacturing recombinant proteins, small-molecule secondary metabolites of medicinal activity. Moreover, it is applied to regenerate agricultural or ornamental plants [10,11,12,13,14,15,16]. Thus, the pressure exerted on plants in arable areas could be relieved by using in vitro established callus culture to obtain the required compounds [10,12]. Tanshinones and other bioactive components present in medicinal plants belong to secondary metabolites. Their abundance in plant tissues is usually significantly lower as compared to the products of primary metabolism. However, such proportions could be improved after plant elicitation, i.e., the exposition to abiotic or biotic stresses or small molecules participating in response to these conditions [17,18,19,20,21,22]. Some elicitors can be applied in combinations to exploit possible synergistic effects [19,21,23,24]. Elicitors commonly applied in plant tissue culture are methyl jasmonate, salicylic acid, yeast extract, abscisic acid, silver ions, and ethylene [17,18,19,20,21,22,23,24]. Yeast extract efficiently increased tanshinone concentration in *S. miltiorrhiza* callus cultures growing in suspension [21]. However, the ethephon was not an effective elicitor in *Rubia cardifolia* callus cell cultures [24]. Plant development, growth, and stress responses are regulated by ethylene, a gaseous phytohormone, via a well-characterized signaling pathway, including a family of ethylene receptors in the membrane of the endoplasmic reticulum (ER), a protein kinase known as constitutive triple response 1 (CTR1) and an ER-localized transmembrane protein of unknown activity, called ethylene-insensitive 2 (EIN2). Finally, the transcription factors EIN3 and EIN3-like (EIL), and ethylene response factors (ERFs) modify the expression of target genes [25]. ERFs belong to the AP2/ERF family, represented in *S. miltiorrhiza* by170 Ap2/ERF *trans*-factors [26,27]. These include Sm128 and Sm152, which positively affect tanshinone concentration [26,27]. Other ERFs regulating tanshinone biosynthesis are SmERF1L1, SmERF6,8, and 73 [28,29,30,31,32]. ERFs bind to the AGCCGCC (GCC box) motif in gene regulatory regions [28,29,30,31,32]. Ethylene could efficiently increase secondary metabolite accumulation in plants [33,34]. Moreover, ethylene applied in the form of gas-releasing factor ethephon (ET) increases tanshinone concentration in *S. miltiorrhiza* [35].

Another factor used as an elicitor to increase the concentration of active ingredients in plant cultures is yeast extract (YE), resembling the plant exposition to fungal pathogens [36]. YE applied in concentration 0.1–1.0% for seven days was found to increase total tanshinone (Tt) concentration from 0.001% to 0.096% in hairy root cultures of *S. miltiorrhiza* transformed with *Rhizobium rhizogenes* ATCC 15,834 [37]. In addition, seven-day elicitation with YE 50, 100, 200, and 300 mg L^−1^ improved dry mass and cryptotanshinone (CT), dihydrotanshinone (DHT), and tanshinone IIA (TIIA) concentration in transformed hairy roots of *Salvia. castanea* Diels *f. tomentosa* Stib [38]. Additionally, the Tt content in *S. miltiorrhiza* hairy roots overexpressing *SmHMGR* (*3-hydroxy-3-methylglutaryl CoA reductase*) and *SmDXR*(*1-deoxy-D-xylulose 5-phosphate reductoisomerase)* may be increased by YE elicitation. The Tt content in transformed hairy roots increased from 3.25 mg g^−1^ DW to 5.86 mg g^−1^ DW after nine-day elicitation with 100 mg L^−1^ YE [23]. However, there is insufficient research addressing the influence of ethylene, YE, or ethylene + YE on the concentration of tanshinones in *S. miltiorrhiza* callus cultures, particularly for relatively long exposure times, i.e., up to 60 days.

YE contains chitin, chitin-derived oligosacharides, peptides, or other biomolecules, mimicking the plant exposure to fungi through the activation of plant pattern recognition receptors (PPR) [36,39]. However, the activation of PPR results in the stimulation of numerous metabolic routes, with pivotal roles played by salicylic acid (SA) and ethylene/jasmonate-dependent signaling pathways [40]. Usually, the SA-mediated response occurs during the colonization by biotrophic pathogens, while the ethylene/jasmonate route regulates the plant response to necrotrophic pathogens [40].

SA exerts its activity through WRKY *trans*-factors, recognizing TTGAC(C/T) *cis*-active elements known as W-boxes in the promoter regions of defense-related genes [41,42,43,44]. Overexpression of *SmWRKY* 1 and 2 increases the tanshinone concentration in *S. miltiorrhiza* [45,46] and influences NPR1 (non-expression of pathogenesis-related genes 1), a critical protein in the SA-regulatory pathway, which is deprived of its DNA binding domain [47]. Therefore, NPR1 associates with bZIP *trans*-factors to use their DNA-binding domains and recognize such SA-responsive TGACG transcription factor binding sites (TFBSs) [41,42,48].

Such coordinated activity by transcription factors in response to stress conditions ameliorates the expression of numerous target genes [40,49]. Although the post-transcriptional stages of gene expression or the activity of the synthesized protein are precisely regulated, the main outcome of *trans*-factor activation is the change in plant secondary metabolism, adapting plants to changes in their habitat [50,51,52]. Therefore, the biosynthesis of such secondary metabolites as tanshinones is increased in response to elicitation [19,20,21,38]. The diterpene tanshinones are synthesized from two five-carbon precursors: isopentenyl diphosphate (IDP) and dimethylallyl diphosphate (DMADP) [53,54,55,56]. These precursors are produced by the cytosolic mevalonic acid (MVA), plastidial methylerythritol-4-phosphate (MEP) pathway, and some exchange of IDP occurs between both cellular compartments [56]. However, the IDP and DMADP produced in *S. miltiorrhiza* by cytoplasm-localized MVA are used mainly to support vegetative growth, while the MEP pathway produces precursors for the secondary metabolism, such as tanshinone diterpene components [55]. The IDP and DMADP moiety is used for the stepwise biosynthesis of a 20-carbon geranylgeranyl diphosphate (GGPP), which then serves as a substrate for copalyl diphosphate synthase 1 and 2, producing (+)-copalyl diphosphate [19,57,58,59]. The (+)-copalyl diphosphate is transformed into miltiradiene, the first committed tanshinone biosynthesis pathway intermediate, by the KSL [20,57,58,59].

Analysis of *SmKSL* genes suggests that the *SmKSL* and *SmKSL1* are associated with tanshinone biosynthesis, while the *SmKSL2* produces *ent*-kaurene, used in gibberellin assembly [20,59]. In total, nine *SmKSL* gene homologs have been identified in *S. miltiorrhiza* [60]. The regulatory region of the *SmKSL1* gene has been isolated and analyzed to reveal the GCCelements participating in response to ethylene [29]. Additionally, Sallaud et al. (2012) isolated a partial sequence of the *KSL* gene and its promoter from *Nicotianatabacum* [61]. The obtained gene encodes a KSL protein, known as NtABS, that uses 8-hydroxy-copalyl diphosphate to produce Z-abienol in the root glandular trichome [61].

To fill the gap related to the regulatory regions of *KSL* genes in *S. miltiorrhiza*, the 5′ regulatory segments of *S. miltiorrhiza* kaurene synthase, known also as *SmKSL* (EF635966.2), were cloned. The obtained promoter sequence was deposited in GenBank at accession number KT899977.1. Initial bioinformatics analysis revealed numerous *cis*-active elements, including W-boxes, TGACG elements, and a core region of ethylene-responsive GCC motifs.

Co-expression studies based on microarrays and transcriptomic results were applied to validate the observed *cis*-active elements. The role of YE and ethylene in the regulation of *SmKSL* was evaluated by RT-PCR. Moreover, YE and ethylene were found to increase the tanshinone concentration in *S. miltiorrhiza* callus cultures maintained for up to 60 days. Simultaneous application of YE and ethylene, in the form of ethephon, was used to verify if they could exert a cooperative or mutually antagonistic effect on *SmKSL* gene expression and tanshinone biosynthesis rate [40].

## 2. Material and Methods

### 2.1. Plant Cultivation

Seeds used for *S. miltiorrhiza* Bunge cultivation were received from the Medicinal Garden of the Department of Pharmacognosy at the Faculty of Pharmacy, Medical University of Łódź (Poland). The development of the plants was maintained on composite soil, in 0.5 L pots (diameter 12 cm), at 26 ± 2 °C under natural light. Eight-week-old plants were used for experiments.

### 2.2. Isolation of Genomic DNA

Genomic DNA was isolated according to Khan et al. (2007) [62]. Approximately 0.75 g of plant material was used for isolation. The genomic DNA concentration and purity were determined based on A_260/280_ and A_260/230_ using a P300 Nanophotometer (Implen, Munich, Germany).

### 2.3. Promoter Isolation and In Silico Characterization

The suspected promoter region of the *S. miltiorrhiza KLS* gene (*SmKSL*) was isolated using a Genome Walker^TM^ Universal kit (Takara Bio USA, Mountain View, CA, USA). Two specific primers, GSP1 5′TGATCATCGGAAATCCTCGGACTGTAACAT3′ and GSP2 5′TTCTCTCCTGCTCCGAGCCCCGTT3′were designed using the 5′ terminal fragments of the *SmKSL* cDNA (GenBank EF635966.2). Primer design was performed according to the Genome Walker^TM^ Universal kit manufacturer guide and previous experiments [63]. OligoCalc server was applied to design gene-specific GSP1 and GSP2, as well as the DNA sequencing primers [64]. The salt adjusted (50 mM NaCl) ^TM^ was used to determine the T_M_ of all PCR primers [64]. Details of the genome walking procedure and the cloning of promoter region are described in earlier research [63].

The alkaline lysis miniprep and phenol-chloroform extraction were used to obtain purified plasmid DNA [65]. Plasmid DNA was sequenced as an external service by the CoreLab facility at the Medical University of Łódź (Poland). The obtained DNA sequence was deposited at GenBank (KT899977.1) and analyzed by PlantPAN3.0 to find potential *cis*-active sequences, tandem repeats, miRNA binding sites, and CpG/CpNpG islands [66]. TATA-box and transcription initiation sites (TIS) were localized using TSSP software and RegSite Plant DB (Softberry Inc., Mount Kisco, NY, USA) [67,68].

### 2.4. Microarray and NGS Co-Expression Data Analysis

The results of previous in silico studies on the *SmKSL* promoter were validated by comparison with *trans*-factors co-expressed with *Arabidopsis thaliana* (*A. thaliana*) *KSL* gene (AT1G79460; *AtKSL*). The *A. thaliana* gene that is homologous to *SmKSL* was identified using Protein BLAST (NCBI, Bethesda, MD, USA) and MEGA X version 10.2.6 (Pennsylvania State University, State College, PA, USA) to analyze gene coding sequences [69]. The microarray and NGS co-expression studies were performed on the Expression Angler tool and the *Arabidopsis* RNA-seq Database [70,71]. The following dataset compendiums were analyzed: AtGenExpress Hormone and Chemical, AtGenExpress Abiotic Stress, AtGenExpress Pathogen, AtGenExpress Tissue, and AtGenExpress Plus—Extended Tissue. According to Usadel et al. (2009), only values of r within the range of 0.7–1.0 were applied in the study [72]. Obtained results were compared with the in silico data provided by PlantPan 3.0. Information on the detected TFs was obtained from the UniProt database [73]. The possible interactions between *trans*-factors were identified by the BioGRID database version 4.4.201 [74].

### 2.5. Callus Induction and Elicitation by YE and ET

The callus samples used in the study were induced and propagated as described by Szymczyk et al. (2022) [63]. Callus cultures (approximately 1000 mg) were cultured on Murashige and Skoog (MS) basal medium supplemented with 3% sucrose, 1% DifcoBacto agar (Difco Laboratories, Detroit, MI, USA), and either 0.5, 2.0, and 4.0% of YE or 0.05, 0.10, 0.25, 0.50, and 1.0 mM ET (Sigma Aldrich, Poznań, Poland). Moreover, YE 0.5% combined with 0.05, 0.10, 0.25, 0.50, and 1.0 mM ET were used together to check the mutual relationship between YE and ET [40,75]. The callus cultures were maintained for up to 60 days. Every 20 days, a fresh solid medium was provided. The control group was the YE and ET untreated callus. The carbohydrate fraction of YE was prepared by ethanol precipitation as described by Chen et al. (2001) [37]. The obtained carbohydrate fraction of YE and ET stock solution (500 mM) were sterilized with a syringe filter (0.4 μm pore size) under a laminar hood before addition to warm (50 °C), freshly-autoclaved MS medium. The pH of each MS medium before adding elicitors was adjusted to 5.7 ± 0.1 with 1N NaOH or HCl. Following this, the media were autoclaved for 20 min at 121 °C, 105 KPa. The glass and steel forceps used to manipulate calluses were sterilized at 200 °C for one hour before entering the laminar hood. ET is recognized as a reliable source of ethylene in plant in vitro cultures [33,76]. ET is absorbed by plant material and used to synthesize ethylene inside the floral tissues [76]. Usually, ET is applied in plant in vitro cultures in concentrations within the range of 0.25–1.5 mM [33]. Potential synergy and antagonism between ET and YE were defined according to Roell et al. (2017) [77]. The quantitative effects observed after the separate application of YE or ET were added to determine the additive effect. Any positive deviation from such additive effect observed after combined YE and ET treatment is defined as synergy and any negative change as antagonism [77].

### 2.6. Calculation of Callus Growth Index

Growth index calculations were performed on callus samples collected after 20, 40, and 60 days. All the experiments were repeated three times, with three replicates per harvesting. Callus untreated with YE or ET was used as a control group. The GI of callus fresh weight was calculated using the following equation provided by Godoy-Hernández and Vázquez-Flota (2006) [78]:GI_F_= (FW_F_− FW_I_)/FW_I_
where GI_F_= GI of callus fresh weight; FW_F_ = final callus fresh weight FW_I_ = initial callus fresh weight.

### 2.7. Tanshinone Extraction Procedure

The tanshinone extraction procedure from callus samples was performed according to previous research [63,79].

### 2.8. UHPLC Analysis

Details of the UHPLC analysis are presented previously [80]. TI, TIIA, CT, and DHT were provided as standard HPLC-grade substances by Sigma Aldrich Poland (Poznań, Poland). Methanol, acetonitrile, and water for UHPLC (J.T. Baker HPLC Analyzed), produced by J.T. Baker (Phillipsburg, New Jersey, USA) were received from the Polish distributor Avantor Performance Materials (Gliwice, Poland).

### 2.9. RNA Isolation and cDNA Synthesis

The RNA was prepared from *S. miltiorrhiza* callus using an Isolate Plant II RNA kit (Bioline, Singapore) (Meridian Bioscience, Memphis, TN, USA) according to the manufacturer’s instructions. RNA was obtained from 80–100 mg samples of plant leaves that were cut off and frozen directly in liquid nitrogen. Potential genomic DNA contamination was removed by RNase-free DNaseI (4 U/sample) digestion. The expected result of DNaseI digestion was confirmed by the quantitative, real-time PCR reaction using control samples of RNA, that were not converted into cDNA. All RNA samples were prepared in triplicate and processed instantly or stored at −80 °C until analysis. The concentration and purity of the prepared RNA were evaluated using a p300 Nanophotometer (Implen, Munich, Germany). The obtained RNA samples indicated an A_260/280_ ratio within the range of 1.6–1.8. The prepared RNA was used to synthesize cDNA using a High-Capacity cDNA Reverse Transcription Kit (Applied Biosystems; Thermo Fisher Scientific, Waltham, MA, USA). The reaction mixture consisted of the following components: dNTPs (4 mM final), anchored oligo (dT)_18_ (1 μM final), 2 μL of 10× buffer, RNase inhibitor (20 U), and a MultiScribe™ Reverse Transcriptase (50 U). The concentration of RNA was adjusted to the value of 0.01 μg/μL in a final volume of 20 μL.

### 2.10. Real-Time PCR 

The relative concentrations of *SmKSL* and *ubiquitin* mRNA in *S. miltiorrhiza* callus samples were analyzed by quantitative real-time PCR. Experiments were performed using a CFX Connect Real-Time PCR Detection System (Bio-Rad, Hercules, CA, USA). TheiTaq™ Universal SYBR^®^ Green Supermix (Bio-Rad Laboratories, Inc, Hercules, CA, USA) was used in the RT-PCR tests. *Ubiquitin* was chosen as a reference gene based on its very stable expression in *S. miltiorrhiza* [81], confirmed by analysis with BestKeeper software, indicating low SD (SD < 1) and relatively high r values [82]. Experiments were performed in duplicate to assure the reproducibility of the method.

The size of the *ubiquitin* gene fragment was found to be 192 bp, as indicated by sequence alignment based on *Populus trichocarpa* cDNA (GenBank FJ438462.1) using primers 5′GTTGATTTTTGCTGGGAAGC3′ (forward) and 5′GATCTTGGCCTTCACGTTGT3′ (reverse) [80,81,82]. Similarly, the *S. miltiorrhiza KSL* gene fragment (GenBank JN831097.1) was found to be 143 bp in length based on the *SmKSL* gene primers 5′GTTTATGAACCCCTAACAATTCA3′ (forward) and 5′TCTTATTGTGGCATTCATTGGTTTT3′ (reverse). Ethidium-bromide agarose gel electrophoresis confirmed the expected size of *Populus trichocarpa ubiquitin-* and *SmKSL*-based PCR products. Samples and negative controls were created in triplicate. Quantitative PCR reactions were performed in separate tubes using the following parameters: initial denaturation (95 °C, 10 min), denaturation (95 °C, 20 s), primer annealing (60 °C, 30 s), extension (72 °C, 20 s). In total, 40 PCR cycles were performed. The reaction mixture contained the following components: 5 μL iTaq™ Universal SYBR^®^ Green supermix (2×), 0.5 μL of each primer, 1 μL of cDNA, and distilled water to a final volume of 10 μL. The relative changes in gene expression were calculated according to Pfaffl [83]. CFX Maestro Software v. 2.2 (Bio-Rad, Hercules, CA, USA) was applied to evaluate the RT-PCR results.

### 2.11. Statistical Analysis

Differences between RT-PCR tested samples were identified based on one-way ANOVA. A *p-*value < 0.05 was considered significant [76]. The Wilcoxon signed-rank test was used to test control samples before and after YE, ET, or YE plus ET treatment to calculate *p* values. The results of the UHPLC and GI analyses were evaluated by the Wilcoxon signed-rank test. Values of *p* < 0.05 were considered statistically significant [63].

## 3. Results

### 3.1. In Silico Analysis of SmKSL Promoter

The obtained DNA fragment is 897 bp long and was deposited in GenBank (KT899977.1). The transcription start site (TSS) and 5′UTR were identified using TSSP software (Figure 1). The coding sequence consists of 25 bp, which is identical to *S. miltiorrhiza* kaurene synthase (*KSL*) (GenBank EF635966.2) (Figure 1). The in silico search revealed a lack of tandem repeats and CpG/CpNpG islands. No TATA-box, in the form of cTATAA/TAT/AA or TCACTATATATAG, was observed within 25–35 bp in the 5′ direction from the TSS. Therefore, the isolated *SmKSL* fragment belongs to the majority of plant TATA-less promoters [67,68,84,85].

The analysis indicated numerous possible binding sites for *trans*-factors that could be homologous to binding sites for ERF, RAV1, and C2H2 *trans*-factors controlling the response to ethylene in *S. miltiorrhiza*. Additionally, potential *trans*-factors related to plant defense, such as WRKY and ANAC2 were noted. The prepared promoter may be regulated by light through the Dof and GATA *trans*-factors. Other processes may be regulated by the following *trans*-factors: TCX6 (DNA methylation maintenance), anthocyanin accumulation (Myb/SANT), and bZIP (plant morphogenesis) (Figure 1). A complete list of putative *cis*-active elements found within obtained promoter by PlantPAN3.0 is presented in Appendix A.

The importance of ERFs to the *SmKSL* gene regulation suggests the presence of GCCGCC in position 846–862 (Figure 1). Although the formal GCC box (AGCCGCC) box was not observed, the six internal nucleotides GCCGCC were found, which are of crucial importance for ERF binding [29].This hexamer is tandemly repeated twice in the 5′UTR of *SmKSL* (Figure 1).

### 3.2. Microarray and NGS Co-Expression Data Analysis

The closest *A. thaliana* homolog to *SmKSL* is *A. thaliana* AT1G79460. The transcription factors and other proteins co-expressed with AT1G79460 were identified by Expression Angler software (Appendix A). Seventeen *trans*-factor genes were found to be co-expressed with AT1G79460 within the r range of 0.7–1.0. These were identified in AtGenExpress Elicitors and AtGenExpress Abiotic Stress (Appendix A). Among them, only three were found among the *A. thaliana* homologs that could bind to the *SmKSL* promoter: NF-YC12, ATHB5, and ARR13 (Appendix A). They serve as general *trans*-factors (NF-YC12) or play specialized roles in response to abscisic acid (ATHB5) or auxin (ARR13) (Appendix A).

A detailed transcriptomic analysis of genes that could be co-expressed with the *A. thaliana* homolog to *SmKSL* indicated an approximate maximum WGCNA correlation score of 0.06 [71].

However, an analysis of available *A. thaliana* transcriptomic studies revealed responses to numerous pathogens as well as abiotic stress conditions (Appendix A) [71]. The participation of ERF and WRKY *trans*-factors in response to biotic stress conditions suggests that they may regulate *SmKSL* promoter activity, as identified by initial in silico analyses [40]. Moreover, previous transcriptomic studies on *SmKSL1* suggest responsiveness to YE [58]. Additionally, four SmERFs (6, 8, 73, and 128) positively regulate *SmKSL1* activity assayed by RT-PCR [27,29,30,31].

### 3.3. RT-PCR Analysis of SmKSL Promoter Activity after YE and ET

RT-PCR analysis revealed decreased *SmKSL* promoter activity after YE 0.5% (24 h), YE 2% (72 h), and YE 4% (72 h). However, YE generally increased activity at the other analyzed time points and YE concentrations. After 48 h, YE increased *SmKSL* gene expression for all applied YE concentrations: 0.5 (1.93-fold), 2 (2.79-fold), and 4% (3.85-fold) (Figure 2).

The reaction of *SmKSL* to ET treatment is clearly concentration and time-dependent. The lowest ET concentration, 0.05 mM, resulted in a 2.34-fold (24 h), a 2.37-fold (48 h) activation, and a neutral 1.02-fold (72 h) effect. The 0.10 mM ET concentration inhibits *SmKSL* expression after 24 h (0.12-fold) but shows activation after 48 (12.73-fold) and 72 h (1.29-fold). The higher ET concentrations (0.25 and 0.50 mM) inhibit *SmKSL* activity for longer: 1.79-fold activation after 48 hours and 3.63-fold after 72 h (Figure 2).

All combinations of 0.5% YE and ET (0.05, 0.10, 0.25, and 0.50 mM) demonstrate *SmKSL* inhibition after 24 h. After 48 hours, a weak inhibitory effect (0.87-fold) was observed only for the highest ET concentration (0.50 mM) combined with the YE 0.5%. Moreover, after a 72 h treatment, *SmKSL* expression (0.26-fold) was only inhibited for ET 0.10 mM plus YE 0.5%, suggesting a dominant activatory role of both compounds, similar to the 48 h time point (Figure 2). After 48 h, the strongest mean activation of *SmKSL* gene expression was noted for ET 0.10 mM plus YE 0.5% (25.85-fold) and ET 0.25 mM plus YE 0.5% (9.12-fold) (Figure 2). Quantitative analysis suggests that YE and ET have dominant antagonistic effects on *SmKSL* gene expression (Appendix A).

### 3.4. Growth Index Rates of Callus Treated by YE, ET and YE Combined with ET

The *S. miltiorrhiza* callus GI was inhibited by YE and ET in a concentration-dependent manner at all tested time points (20, 40, and 60 days). For example, at 60 days, the callus growing on 0.5, 2, and 4% YE demonstrated the following respective GI values: 4.52 ± 0.28, 1.67 ± 0.26, and 0.52 ± 0.10 (Figure 3). The GI for control samples was 28.55 ± 2.16. As the callus growth rate was better on YE 0.5% than higher YE concentrations, YE 0.5% was chosen for further UPLC analysis and testing in combination with the four tested ET concentrations.

The application of ET at 0.05, 0.10, 0.25, and 0.50 mM also retarded the *S. miltiorrhiza* callus growth rate compared to controls (Figure 3). At 60 days, the mean GI values were 8.15 ± 1.12, 3.72 ± 0.43, 2.82 ± 0.30, and 1.61 ± 0.17observed for ET 0.05, 0.10, 0.25, and 0.50 mM respectively. The GI for control samples was 30.14 ± 4.81. The combination of YE 0.5% and ET for 60 days strongly retards the callus growth rate as compared to the control, with mean GI values of 1.61 ± 0.19, 1.48 ± 0.22, 1.31 ± 0.12, and 1.16 ± 0.21 for ET 0.05, 0,10, 0.25, and 0.50 mM, respectively (Figure 3). The GI for control samples was 30.49 ± 3.12. The differences between control and tested values were statistically significant according to the Wilcoxon signed-rank test (*p*<0.01).

### 3.5. Induction of Tanshione Biosynthesis by YE, ET and YE Combined with ET

The elicitation results suggest that the most significant rise in tanshinone concentration occurred after YE treatment, and the use of ET or YE + ET was less efficient. Quantitative analysis indicated that among the three tested YE levels, 0.5% had the greatest effect on the tanshinone concentration (Figure 4). Increased Tt concentration was noted after 50 days of elicitation for all three tested YE concentrations (Figure 4), with 0.5% YE yielding the highest Tt value (13.30 ± 1.09 mg g^−1^ DW), followed by YE 2% (7.66 ± 0.16 mg g^−1^ DW) and YE 4% (5.52 ± 0.12 mg g^−1^ DW) (Figure 4).

Changes in tanshinone concentration are consistent with the kinetics observed for Tt. The highest values of CT, DHT, and TIIA were noted after 50 days of YE induction, and the most efficient stimulation was provided by 0.5% YE (Figure 4). Results obtained for YE 0.5% suggest that CT dominates, followed by DHT and then TIIA. TI was not induced by YE at any applied concentration (Figure 4). The approximate ratio of CT, DHT, and TIIA did not significantly differ between YE concentrations, and the approximate proportions calculated for all samples were 81.42% (CT), 17.06% (DHT), and 1.52% (TIIA).

ET was a very weak inducer of Tt concentration. However, some regularities could be observed in the ET elicitation activity. The lowest ET concentration (0.05 mM) only increased the mean Tt concentration to 0.038 mg g^−1^ DW after 10 days of elicitation, while longer exposure decreased the Tt concentration (Figure 5). The application of 0.10 mM ET elevated the Tt level up to the mean of 0.070 ± 0.003 mg g^−1^ DW after 40 days and 0.06 ± 0.01 mg g^−1^ DW after 50 days. The highest mean level of Tt (0.42 mg ± 0.01 mg g^−1^ DW) was noted after 60 days of exposure to 0.10 mM ET. The higher ET level (0.25 mM) augmented Tt relatively quickly: the mean Tt was 0.15 ± 0.01 mg g^−1^ DW after 10 days and further growth of Tt required more time, whichis similar to the effect of 0.05 mM ET. The highest applied ET concentration (0.50 mM) increased Tt sharply to a maximum level of 0.67 ± 0.04 mg g^−1^ DW mg after 20 days. Longer exposure to 0.50 mM ET gradually decreased Tt concentration to 0.26 ± 0.01 mg g^−1^ DW.

The kinetics of the CT, DHT, TI, and TIIA changes are closely related to those noted for Tt (Figure 5). Among these, CT dominated (51.21%), followed by DHT (45.00%). The level of TIIA (3.79%) is much lower and TI was not detected. These relative concentrations of DHT and CT seem to be specific for ET induction, with YE induction results in a strong prevalence of CT over DHT.

For Tt induction by 0.5% YE combined with ET 0.05, 0.10, 0.25, and 0.50 mM, 0.5% YE appears to have a greater influence than ET (Figure 6). Indeed, YE plus ET yields higher Tt, CT, DHT, and TIIA values than ET alone but lower than YE alone. Moreover, the kinetics of Tt growth shows that after the application of 0.5% YE combined with 0.05 (8.20 ± 0.21 mg g^−1^ DW) and 0.10 mM ET (4.33 ± 0.26 mg g^−1^ DW), the highest Tt concentration was reached after 50 days; however, both values are much lower than YE alone (Figure 6). The peak value of Tt after 0.5% YE combined with 0.25 mM ET (1.32 ± 0.21 mg g^−1^ DW) was observed relatively quickly after 20 days of elicitation. Moreover, the highest Tt level after 0.5% YE and 0.50 mM ET was noted after 40 days (0.159 ± 0.003 mg g^−1^ DW) (Figure 6). The mean percentage values for tanshinones are 73.77% (CT), 23.17% (DHT), and 3.06% (TIIA), resembling the ratio observed after YE treatment. The interaction between the YE and ET seems to be antagonistic as the Tt concentration noted after the combined action of YE and ET is lower than the sum of Tt concentrations for YE and ET acting separately (Appendix A).

No Tt, CT, DHT, TI, or TIIA were observed in the control samples. Differences between control (0) and non-zero tested samples were statistically significant according to the Wilcoxon signed-rank test at *p* < 0.01. Moreover, a comparison of Tt concentration noted for YE 0.5% and YE 2% as well as YE 0.5% and YE 4%, indicating that they are significantly different (*p* < 0.01) according to the Wilcoxon signed-rank test for all analyzed timepoints (10–60 d). A similar analysis performed for the following pairs; ET 0.05 mM-ET 0.10 mM, ET 0.05 mM-ET 0.25 mM, and ET 0.05 mM-ET 0.50 mM indicated the statistically significant differences of Tt concentration (*p* < 0.01) for all analyzed time-points 10-60 d. The same was true for subsequent pairs; YE 0.5% + ET 0.05 mM-YE 0.5% + ET 0.10 mM, YE 0.5% + ET 0.05 mM-YE 0.5% + ET 0.25 mM and YE 0.5% + ET 0.05 mM-YE 0.5% + ET 0.50 mM. Additionally, it was here statistically significant differences in Tt concentration (*p* < 0.01) for all tested timepoints 10–60 d were observed

## 4. Discussion

The presented research analyses the *SmKSL* gene promoter to identify putative *cis*-active elements and their role in gene expression and responsiveness to ET, YE, and ET+YE. The isolated 897 bp long DNA fragment consists of a promoter (763 bp), 5’UTR (109 bp), and a short 25 bp CDS sequence. The isolated fragment does not contain a TATA-box and belongs to the more numerous TATA-less promoter population, representing 71% of all promotors in *A. thaliana* [84,85]. Additionally, the polypyrimidine tract, the signal for organizing spliceosomal complexes, was not observed in the isolated promoter [86,87]. The role of epigenetic factors in the regulation of *SmKSL* is low due to the absence of CpG/CpNpG islands [88]. However, plants are able to methylate other DNA elements and not only the cytosines present in CG dinucleotides but also those in all other potential sequence contexts at CHG and CHH positions (where H = A, T, or C) [89].

Therefore, such a conclusion should be taken guardedly. Isolated *SmKSL* promoter contains regulatory *cis-*active elements believed to be associated with the response to light (Dof and GATA), methylation maintenance (TCX6), plant morphogenesis (bZIP), and anthocyanin biosynthesis (Myb/SANT). However, particular attention was put on the defense processes induced by YE and ethylene. Such interest was triggered by two factors, the first is the detection of WRKYs and ANAC2 *cis*-active elements, responsible for plant-defence induction, empowered by the presence of a tandemly-repeated core element GCCGCC of the GCC box (AGCCGCC), which could participate in a reaction to ethylene. Plant exposure to the fungal components present in YE could trigger the salicylic acid-mediated response typical of biotrophic pathogen invasion, which finally leads to the activation of WRKYs or TGACG-binding *trans*-factors [40]. However, ethylene and ERFs participate more in the plant response to necrotrophic pathogens [40]. The relationship between ET and YE is not clearly described, particularly in *S. miltiorrhiza* callus cultures lasting up to 60 days. The results of previous studies suggest that this relationship could be antagonistic [40].

Attempts to validate *trans*-factors responding to ET and YE through co-expression and transcriptomic studies were not effective. However, the available results of RNA sequencing and RT-PCR realized on a close homolog of *SmKSL* known as *SmKSL1* indicated possible responsiveness to YE [58]. Moreover, RT-PCR findings suggest that four *Sm*ERFs (6,8,73, and 128) positively regulate *SmKSL1* activity [27,29,30,31]. Additionally, ET (200 μg/L) added to *S. miltiorrhiza* hairy root cultures increased the *SmKSL1* gene expression after two hours, eight hours, and eight days [35]. RT-PCR experiments were performed to verify the responsiveness of *SmKSL* to YE, and ET treatment suggests the dominant activatory role of YE 0.5, 2, and 4% after 48 h. The lowest ET concentration, 0.05 mM, shows activation of *SmKSL* after 24 and 48 h. However, higher ET concentrations show initial inhibition of gene transcription, with a duration proportional to ET concentration. While 0.10 mM ET inhibits *SmKSL* expression after 24 h and activates after 48 and 72 h, 0.25 and 0.50 mM ET block *SmKSL* expression after 24 and 48 h and activates it after 72 h. When applied simultaneously, YE and ET act antagonistically on *SmKSL* gene expression.

Although YE and ET activate the *SmKSL* gene expression, it is not clear if the same effect occurs in relation to other genes encoding enzymes from MEP and MVA pathways or later stages of tanshinone biosynthesis. Such induction could stimulate the expression of such genes and the level of corresponding enzymes, resulting in an increasing tanshinone biosynthesis rate. The outcome of previous studies shows the presence of W-boxes and SA-responsive *cis*-elements in the proximal promoters of ten (28.57%) among 35 tested *A. thaliana* MEP and MVA pathway genes [63,90]. In addition, three out ofseven (42.86%) *S. miltiorrhiza* genes participating in tanshinone precursor biosynthesis also contain these *cis*-active motifs, which may explain the increased tanshinone concentration in response to YE treatment observed in the present study [63,90]. Moreover, no ET-responsive complete GCC-box (AGCCGCC) was found in the proximal promoters of genes from the MVA, MEP, and tanshinone biosynthesis pathways [63]. These data may explain a much weaker metabolic response to ET treatment as compared to YE. Additionally, the results of RT-PCR, RNA sequencing, or proteomic studies suggest that YE or polysaccharide fractions from endophytic fungi activate numerous genes of crucial importance for tanshinone precursor biosynthesis, leading to elevated tanshinone concentration [91,92].

Although the activatory role of YE and ET on tanshinone biosynthesis in Danshen hairy roots was observed for up to 30 days, it is not clear how these factors influence tanshinone biosynthesis in *S. miltiorrhiza* callus culture maintained for a longer time [35,37,38,93]. Additionally, the influence of combined YE and ET on tanshinone biosynthesis is not clear. UHPLC findings suggest that the YE-dependent accumulation of Tt concentration in *S. miltiorrhiza* callus cultures progresses rather slowly, showing the highest value after 50 days of elicitation. A generally indolent increase of tanshinone concentration in hairy root culture was noted in previous studies [35,37,38,93]. Similarly, the Tt concentration reached 0.22 ± 0.373mgg^−1^ DW after 50 days of 0.4 mM SA elicitation [90]. However, the early peak of Tt concentration (0.08%) occurred after 20 days of 100 µM MeJa induction [63]. The most efficient elicitor activity indicated 0.5% YE (13.30 ± 1.09 mg g^−1^ DW), while the higher YE concentrations 2 (7.66 ± 0.16 mg g^−1^ DW) and 4% (5.52 ± 0.12 mg g^−1^ DW) seems to be less efficient. The superiority of 0.5% YE as a booster of tanshinone biosynthesis was observed by Chen et al., 2001 [37].The obtained tanshinone values in callus culture are rather high, as the Ti-transformed suspension cultures of *S. miltiorrhiza* indicated 0.06–0.08% (Chen et al. 1997): a significantly lower value than in roots regenerated in vitro (0.269–1.137%) or native plant roots growing in China (0.260–0.388%) [9,18,19,94]. Although the observed concentrations of tanshinones are high, they are not unprecedented. Wu et. al. (2003) report a mean value of 4.59 mg g^−1^ DW CT after 60 days of 0.2 mg L N^6^-benzyladenine elicitation; Yu et al. (2019) also note approximately 5 mg g^−1^ DW CT after seven-day elicitation with 200 µM SA and even 15.5 mg g^−1^ DW CT and 7 mg g^−1^ DW TIIA after seven days of 60 µM silver nitrate elicitation [9,22].

YE elicitation has been found to strongly inhibit *S. miltiorrhiza* cell culture growth [21,95]. However, other authors indicate that YE has a stimulatory role inthe GI of *S. miltiorrhiza* hairy roots [37,93]. The dominant tanshinones induced in callus culture by YE are CT (81.42%) and DHT (17.06%), which could be explained by their position in the tanshinone biosynthesis pathway, where CT is further modified to TIIA, and DHT serves as a substrate to produce TI [55,56,57,96]. Moreover, CT indicates phytoalexin activity, is concentrated in cells, and is directly involved in the response to yeast extract elicitation [17]. Similar results suggest that DHT and CT predominate were provided by other authors in callus or hairy root cultures [7,19,20,37,93,95]. However, as the sites of tanshinone biosynthesis are localized in roots, TI and TIIA are dominant in the hairy roots of *S. miltiorrhiza*, while CT and DHT are a minority [58].

ET is a much weaker inducer of tanshinone biosynthesis than YE. The highest ET concentration 0.50 mM raised Tt sharply to reach a maximum level of 0.67 ± 0.04 mg g^−1^ DW after 20 days. However, the lower concentration (0.25 mM) required 60 days to reach Tt 0.23 mg ± 0.01 mg g^−1^ DW. The same is true for 0.10 mM ET, which required 60 days to reach max. Tt (0.42 mg ± 0.01 mg g^−1^ DW). A relatively slow accumulation of Tt was noted after ET elicitation. The dominant tanshinone was CT (51.21%) followed by DHT (45.00%). Hence, ET elicitation is a stronger inducer of DHT tanshinone biosynthesis than YE. [96].

The influence of YE 0.5% + 0.05, 0.10, 0.25, or 0.50 mM ET on the Tt level shows their antagonistic relationship. Increasing ET concentration resulted in a falling maximum Tt value: 13.30 mg g^-1^ for DW 0.5% YE alone to 8.20 mg g^−1^ DW (0.5% YE + 0.05 mM ET) and 0.159 mg g^−1^ DW (0.5% YE + ET 0.50 mM). In addition, the proportions of CT and DHT were similar to those observed after stimulation by pure YE, suggesting that YE plays a key role in regulating the tanshinone biosynthesis pathway branches producing CT or DHT.

The GI rates were found to decrease proportionally to elicitor concentration. Such relationships reflect the increasing scale of metabolic shift towards the secondary metabolic pathways, away from the primary ones, and the supply of metabolites for the growth processes [97,98].

Future studies including chromatin immunoprecipitation and yeast one-hybrid screens could experimentally confirm the role of ERFs, WRKYs, or TGACG-binding *trans-*factors in regulating the *SmKSL* promoter activity.

## 5. Conclusions

The present study analyses the promoter region of the *SmKSL*, containing numerous potential *cis*-active sites that could be bound by *trans*-factors responding to YE and ET. Their role in *SmKSL* expression activation was confirmed by RT-PCR. The results of the UHPLC analysis indicate that YE efficiently stimulates tanshinone accumulation in solid callus cultures of *S. miltiorrhiza*. However, this process is rather slow, as the maximum values were reached after 50 days of YE elicitation. A comparison of tanshinone accumulation with changes in GI value suggests that among three applied YE concentrations, 0.5, 2, and 4%, the optimal value is 0.5%. ET is a much weaker inducer of tanshinone concentration in *S. miltiorrhiza* solid callus cultures than YE. Moreover, YE and ET could activate different branches of the tanshinone biosynthesis pathway, as CT is the dominant tanshinone after YE treatment, while the ET elicitation maintains CT and DHT at comparable levels.

## Figures and Tables

**Figure 1 genes-14-00054-f001:**
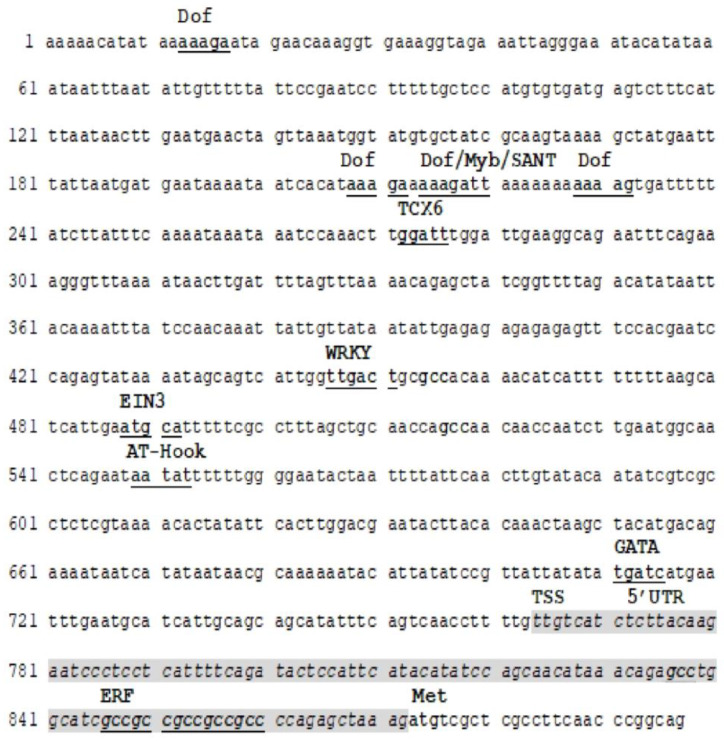
Sequence of *S. miltiorrhiza KSL* promoter region (nt 1–763), 5′UTR (nt 764–872), and 5′ fragment of KSL cDNA (nt 873–897). Only strand + is provided. Positions of *cis*-active elementswere underlined. *SmKSL* gene 5′UTR is shaded and written initalics.

**Figure 2 genes-14-00054-f002:**
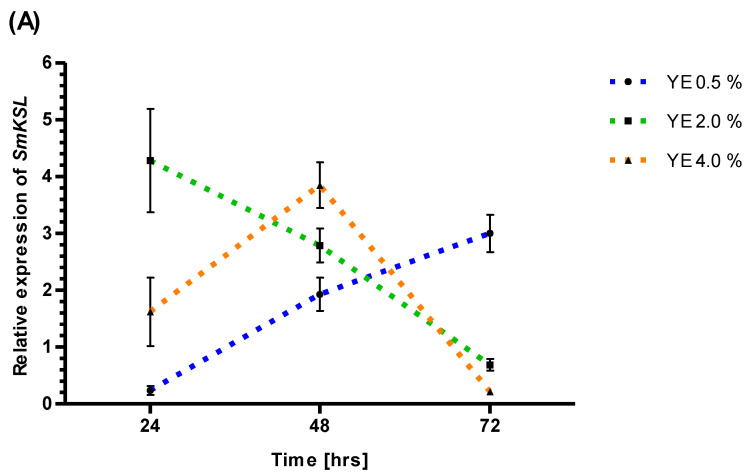
Temporal changes of *SmKSL* gene expression evaluated at 24, 48, and 72 h after treatmentby 0.5, 2.0, 4.0% YE (**A**), 0.05, 0.10, 0.25, 0.50 mM ET (**B**), 0.5% YE plus 0.05, 0.10, 0.25, and 0.50 mM ET (**C**). Results presented as mean +/− standard deviation.

**Figure 3 genes-14-00054-f003:**
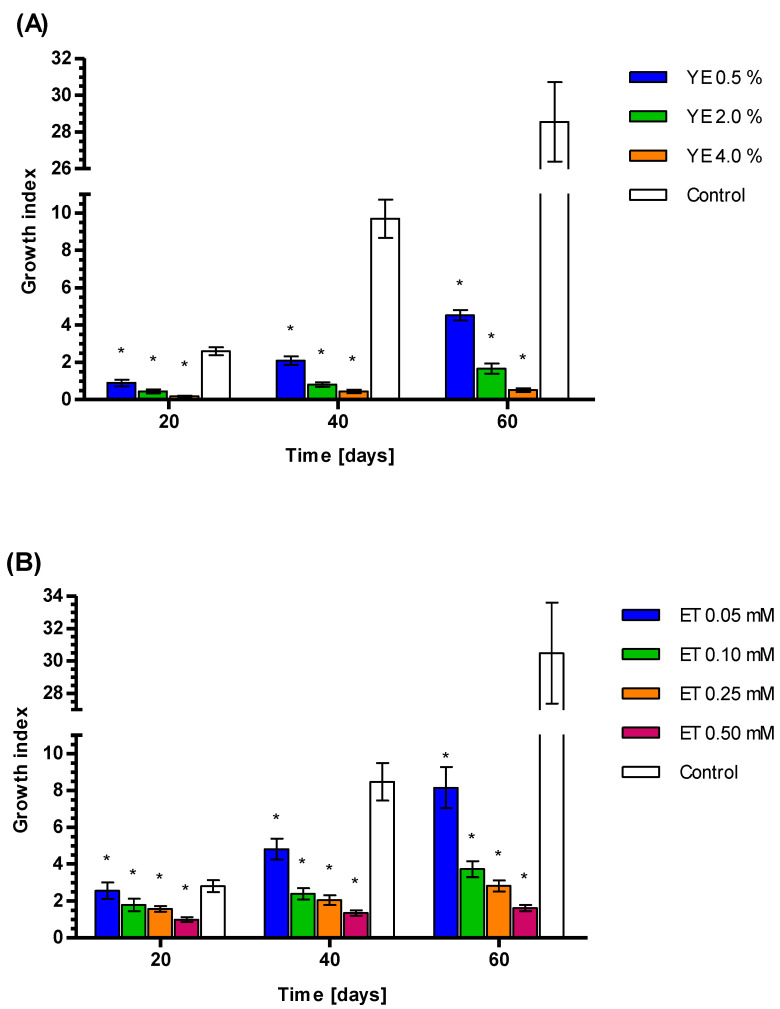
Growth index for fresh *S. miltiorrhiza* callus after treatment by 0.5, 2.0, 4.0% YE (**A**), 0.05, 0.10, 0.25, 0.50 mM ET (**B**), 0.5% YE plus 0.05, 0.10, 0.25, and 0.50 mM ET (**C**). Results presented as mean +/− standard deviation. Differences between control and tested values were statistically significant according to the Wilcoxon signed-rank test at *p* < 0.01. Significant results were marked by an asterisk (*).

**Figure 4 genes-14-00054-f004:**
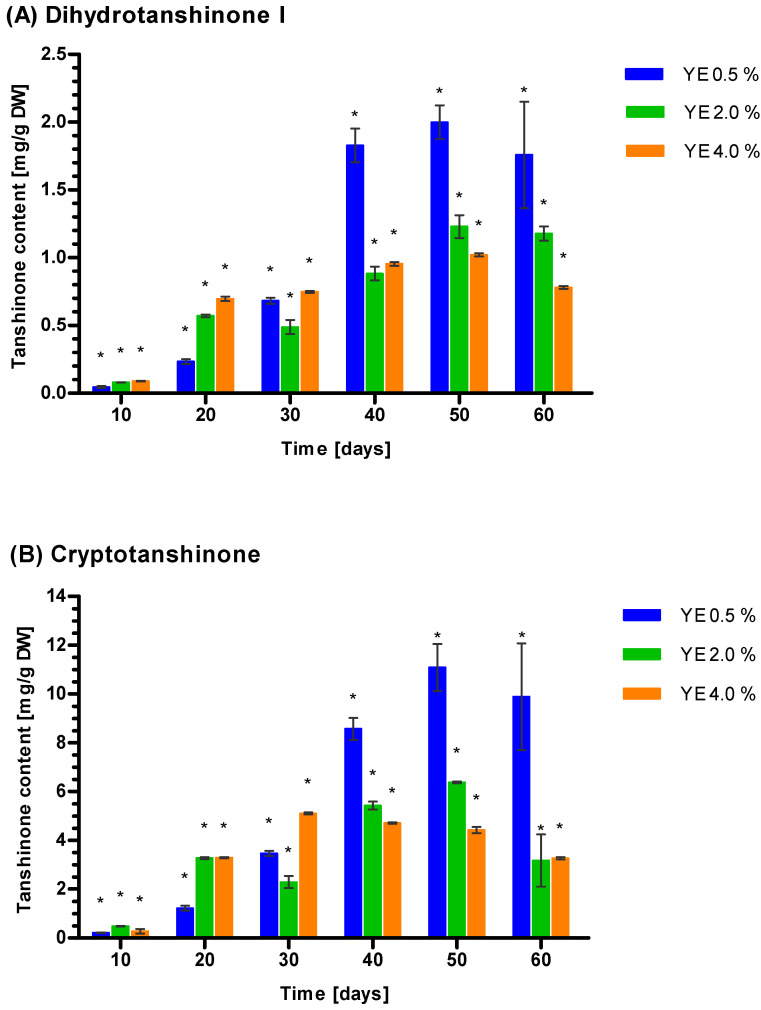
Concentration of three tanshinones dihydrotanshinone I (**A**), cryptotanshinone (**B**), tanshinone IIA (**C**), and total tanshinone (**D**) in *S. miltiorrhiza* callus presented as a function of yeast extract (YE) concentration (0.5, 2.0, and 4.0%) and elicitation time of 10–60 days. Control and tanhinone I values were not detectable. Differences between control (0) and non-zero tested samples were statistically significant according to the Wilcoxon signed-rank test at *p* < 0.01. Significant results were marked by an asterisk (*).

**Figure 5 genes-14-00054-f005:**
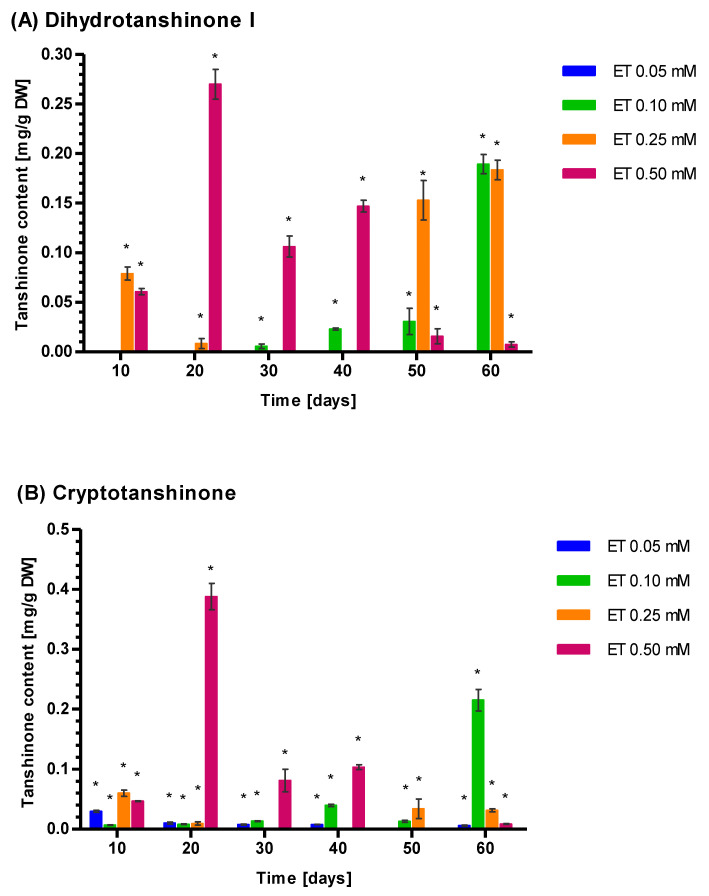
Concentration of three tanshinones dihydrotanshinone I (**A**), cryptotanshinone (**B**), tanshinone IIA (**C**), and total tanshinone (**D**) in *S. miltiorrhiza* callus presented as a function of ethephon(ET) concentration (0.05, 0.10, 0.25, and 0.50 mM) and elicitation time of 10–60 days. Control and tanhinone I values were not detectable. Differences between control (0) and non-zero tested samples were statistically significant according to the Wilcoxon signed-rank test at *p* < 0.01. Significant results were marked by an asterisk (*).

**Figure 6 genes-14-00054-f006:**
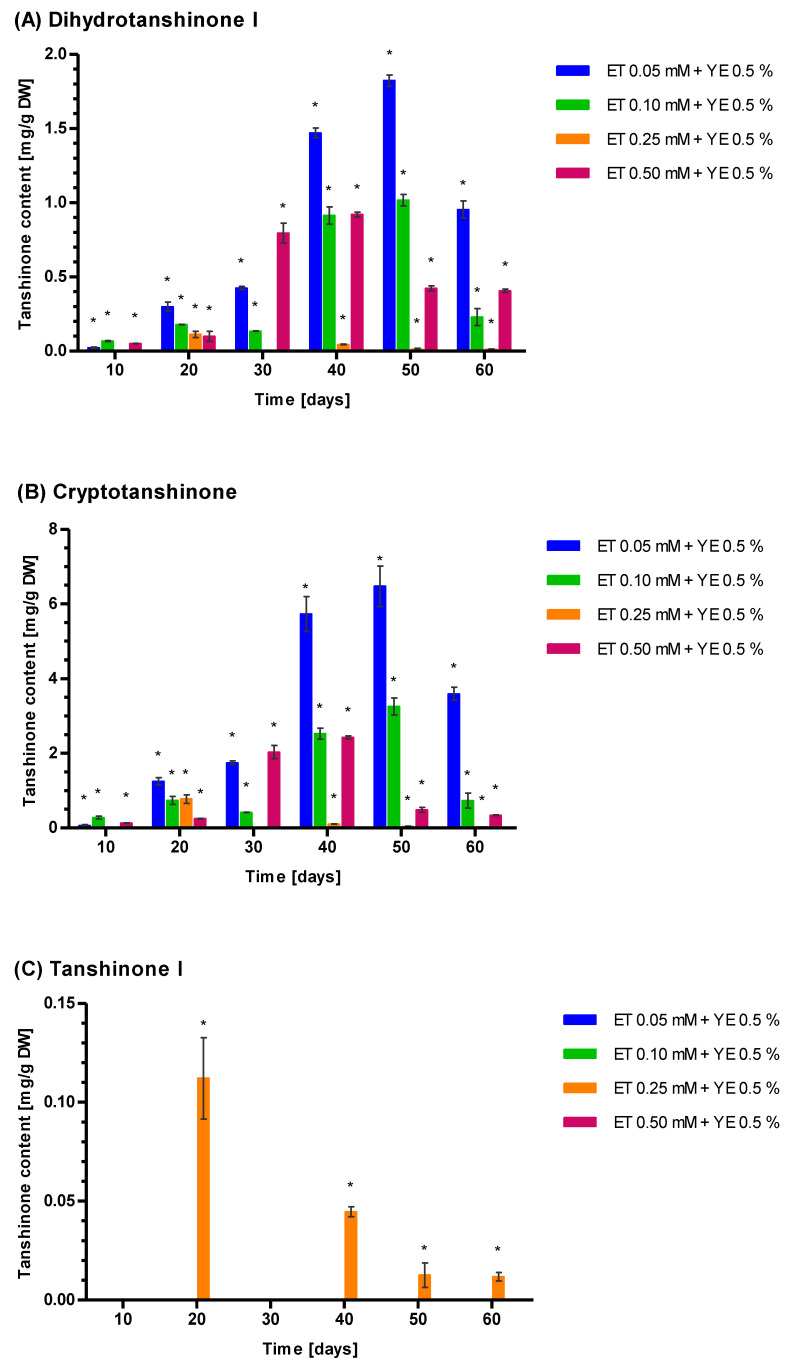
Concentration of four tanshinones dihydrotanshinone I (**A**), cryptotanshinone (**B**), tanshinone I (**C**), tanshinone IIA (**D**), and total tanshinone (**E**) in *S. miltiorrhiza* callus presented as a function of yeast extract (YE) 0.5% combined with ethephon (ET) concentration (0.05, 0.10, 0.25, and 0.50 mM) and the elicitation time of 10–60 days. Control values were not detectable. Differences between control (0) and non-zero tested samples were statistically significant according to the Wilcoxon signed-rank test at *p* < 0.01. Significant results were marked by an asterisk (*).

## Data Availability

Not applicable.

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
