# Peer review of "Isolation of Salvia miltiorrhiza Kaurene Synthase-like (KSL) Gene Promoter and Its Regulation by Ethephon and Yeast Extract"

_genes, 2022, doi:10.3390/genes14010054_

Round 1
Reviewer 1 Report
The manuscript entitled "Isolation of Salvia miltiorrhiza kaurene synthase-like (KSL) gene promoter and its regulation by ethephon and yeast extract" examined the influence of ethylene (ET) and yeast extract (YE) on the expression of SmKSL gene and the regulation of tanshinone biosynthesis. The results confirmed role of YE and ethylene in the regulation of SmKSL and evaluated by RT-PCR. Moreover, the concentration of tanshinone in S. miltiorrhiza callus cultures maintained for up to 60 days found to be increased in presence of YE and ET. In addition, YE and ET could activate different branches of the tanshinone biosynthesis pathway, as the CT is the dominant tanshinone after YE treatment while the ET elicitation maintains CT and DHT at comparable levels.
The title and abstract are appropriate for the content of the text. The introduction is quite thorough and includes an adequate number of recent articles. All the cited references relevant to the research.
Overall, the manuscript represents a generally well-written and well-organized article. Moreover, the experimental design is appropriate, and the results are clearly presented and discussed.
Discussion:
Line 601 – Please change “conformed by” to “confirmed by”
Line 602- Please change “However” to “However,”
Line 603- Please change “stimulatestanshinone” to “stimulates tanshinone”
Line 603- Please change “slow” to “slow,”
Line 604- Please change “Comparison” to “A comparison”
Line 605- Please change “suggest” to “suggests”
Line 607- Please change “However” to “However,”
Line 609- Please change “while” to “ ,while”
Author Response
Responses to Reviewer 1 comments
All Authors are very grateful to the Reviewer nr 1 for the valuable comments.
Following changes were introduced in the Conclusion section nr 7 as suggested by the Reviewer nr 1:
Line 601 – Please change “conformed by” to “confirmed by”
corrected
Line 602- Please change “However” to “However,”
corrected
Line 603- Please change “stimulates tanshinone” to “stimulates tanshinone”
corrected
Line 603- Pleasechange “slow” to “slow,”
corrected
Line 604- Pleasechange “Comparison” to “A comparison”
corrected
Line 605- Pleasechange “suggest” to “suggests”
corrected
Line 607- Pleasechange “However” to “However,”
corrected
Line 609- Pleasechange “while” to “ ,while”
corrected
Also the word „sites” was added to the first sentence to make it more clear.
Reviewer 2 Report
Dear Editor/Authors,
The manuscript examines the influence of ethylene and yeast extract on SmKSL gene expression and tanshinone biosynthesis regulation in Salvia miltiorrhiza callus culture. The results are promising and can be used for commercial application specially for the production of cryptotanshinone using yeast extract as an elicitor. However, the manuscript is having coherence issue and needs a minor revision. In the introduction section add a paragraph about elicitors and highlight the use of ethylene and yeast extract in callus cultures. Rephrase the long sentences in introduction, results and discussion section.
Author Response
Responses to Reviewer 2 comments
All Authors are very grateful to the Reviewer nr 2 for the valuable comments.
Following changes were introduced in the text as suggested by the Reviewer nr 2:
- In the introduction section add a paragraph about elicitors.
Following fragment was addend to the Introduction section:
Tanshinones and other bioactive components present in medicinal plants belongs to secondary metabolites. They abundance in plant tissues is usually significantly lower as compared to the products of primary metabolism. However, such proportions could be improved after plant elicitation, i.e. the exposition to abiotic or biotic stresses or small molecules participating in response to these conditions [17-22].
Elicitors commonly applied in plant tissue culture are methyl jasmonate, salicylic acid, yeast extract, abscisic acid, silver ions and ethylene [17-24].
- In the introduction section highlight the use of ethylene and yeast extract in callus cultures.
Following fragment was addend to the Introduction section:
Yeast extract efficiently increased tanshinone concentration in S. miltiorrhiza callus cultures growing in suspension [21]. However, the ethephon was not an effective elicitor in Rubia cardifolia callus cell cultures [24].
- Rephrase the long sentences in introduction, results and discussion section. Long sentences were rephrased as stated below:
- Following sentence in the introduction section: Although tanshinones are predominantly acquired from the roots of S. miltiorrhiza field plants, this source is becoming steadily impoverished by climate change, soil pollution, water deficit, and incremental stepping of arable area, making traditional production more difficult [6].
is rephrased to:
Tanshinones are predominantly acquired from the roots of S. miltiorrhiza field plants. However, this source is becoming steadily impoverished by climate change, soil pollution, water deficit, and incremental stepping of arable area, ma king traditional production more difficult [6].
II:
Following sentence in the introduction section: Callus culture offers great potential as a tool for manufacturing recombinant proteins, small-molecule secondary metabolites of medicinal activity or regenerating agricultural or ornamental plants [10-16].
is rephrased to:
Callus culture offers great potential as a tool for manufacturing recombinant proteins, small-molecule secondary metabolites of medicinal activity. Moreover, it is applied to regenerate agricultural or ornamental plants [10-16].
Following sentence in the results section: No TATA-box, in the form of cTATAA/TAT/AA or TCACTATATATAG, was observed within 25-35 bp in the 5’ direction from the TSS, suggesting that the isolated SmKSL fragment belongs to the majority of plant TATA-less promoters [67,68,84,85].
is rephrased to:
No TATA-box, in the form of cTATAA/TAT/AA or TCACTATATATAG, was observed within 25-35 bp in the 5’ direction from the TSS. Therefore, the isolated SmKSL fragment belongs to the majority of plant TATA-less promoters [67,68,84,85].
Following sentence in the results section: Seventeen trans-factor genes were found to be co-expressed with AT1G79460 within the r range 0.7-1.0; these were identified in AtGenExpress Elicitors and AtGenExpress Abiotic Stress (Table S2).
is rephrased to: Seventeen trans-factor genes were found to be co-expressed with AT1G79460 within the r range 0.7-1.0. These were identified in AtGenExpress Elicitors and AtGenExpress Abiotic Stress (Table S2).